

# On causality of extreme events

Massimiliano Zanin

Department of Life Sciences, Innaxis Foundation & Research Institute, Madrid, Spain
Departamento de Engenharia Electrotécnica, Universidade Nova de Lisboa, Lisbon, Portugal

## ABSTRACT

Multiple metrics have been developed to detect causality relations between data describing the elements constituting complex systems, all of them considering their evolution through time. Here we propose a metric able to detect causality within static data sets, by analysing how extreme events in one element correspond to the appearance of extreme events in a second one. The metric is able to detect non-linear causalities; to analyse both cross-sectional and longitudinal data sets; and to discriminate between real causalities and correlations caused by confounding factors. We validate the metric through synthetic data, dynamical and chaotic systems, and data representing the human brain activity in a cognitive task. We further show how the proposed metric is able to outperform classical causality metrics, provided non-linear relationships are present and large enough data sets are available.

# INTRODUCTION

Detecting causality relationships between the elements composing a complex system is an old, though unsolved problem (*Pearl*, *2003*; *Pearl*, *2009*). The origin of the concept of *causality* goes back to the ancient Greek phylosophy, according to which causal investigation was the search for an answer to the question "why?" (*Evans*, *1959*; *Hankinson*, *1998*); and the debate was still hot in the late 18th century, in the work of David Hume (*Hume*, *1965*) and his argument that causality cannot be rationally demonstrated.

In the last few decades there has been an increasing interest for the creation of metrics able to detect causality in real data, in order to improve our understanding of systems that cannot directly be described. For instance, while one may suspect that the gross domestic product of a country and its unemployment rate may be related, it is difficult to prove the presence of this relationship, as economical models are neither perfect nor complete. The same happens when one tries to infer if a gene is regulating a second one, in the absence of a complete model of their dynamics, or of a *pathway*. The solution is thus to analyse if the dynamics of these indicators are connected. Among the best known causality metrics, examples include Granger causality, cointegration, or transfer entropy (*Granger*, *1988a*; *Granger*, *1988b*; *Schreiber*, *2000*; *Staniek & Lehnertz*, *2008*; *Verdes*, *2005*), to name a few.

All proposed causality metrics share a common characteristic: causality is defined as a relation existing in the temporal domain, and thus require the study of pairs of time series. For instance, for two processes $\mathcal{X}$ and $\mathcal{Y}$, the transfer entropy is defined as the reduction in the uncertainty about the future of $\mathcal{Y}$ when one includes information about the past of

Corresponding author
Massimiliano Zanin,
mzanin@innaxis.org,
massimiliano.zanin@ctb.upm.es

$\mathcal{X}$ (*Schreiber*, *2000*). Similarly, the Granger causality involves estimating the reduction in the error of an autoregressive linear model of $\mathcal{Y}$ given the history of $\mathcal{X}$ (*Granger*, *1988b*). Associating causality to the temporal domain is intuitive, due to the way the human brain incorporates time into our perception of causality (*Leslie & Keeble*, *1987*; *Tanaka, Balleine & O'Doherty*, *2008*). To exemplify, if we see a ball approaching a window, and just after the window broken, we can safely conclude that the first event was the cause of the second—and thus that causality is a relation between the past and the future. The need of a time evolution is nevertheless an important limiting factor when studying systems whose dynamics through time cannot easily be observed. Consider genetic analysis: one single measurement is usually available per subject and gene, precluding the estimation of gene–gene interactions through a causal analysis solely based on expression levels, as the corresponding time evolution would not be accessible.

When only vectors of observations are available, i.e., vectors representing static observations of different realisations of the same system, it is customary to resort to statistics. This can be classical statistics, for then defining the relationship in terms of linear or non-linear correlations; or Bayesian statistics and the vast field of statistical learning and data mining (*Vapnik*, *2013*; *Zanin et al.*, *2016*). Although correlation, and statistical learning in general, appear *prima facie* as an interesting solution, they present the important drawback of not being able of discriminating between real and spurious causalities. Suppose one is studying a system composed of three interconnected elements, as the one depicted in Fig. 1 (i), with the aim of detecting if the dynamics of element $\mathcal{C}$ is *caused* by $\mathcal{B}$. Additionally, no time series are available, and elements are described through vectors of cross-sectional observations; in other words, multiple realisations of the same system are available, but each one of them can only be observed at a single moment in time. A statistically significant correlation between $\mathcal{B}$ and $\mathcal{C}$ may be found both when a true causality is present (Fig. 1(iii)), and when both elements are driven by an unobserved confounding element $\mathcal{A}$ (Fig. 1(ii)).

In order to tackle the scenario of Fig. 1, in this contribution we propose a novel metric for detecting causality from observational data. It entails three innovative points. First, it is defined on vectors of observation, which do not have to necessarily represent a time evolution. In other words, input vectors may correspond to gene expression levels measured in a population, i.e., to a cross-sectional study; or, but not necessarily, to multiple observations of the same subject, i.e., to a longitudinal study. Second, the method is based on the detection of extreme events, and on their appearance statistics. This is not dissimilar to Granger causality, as the latter measures how shocks in one time series are explained by a second one; but without the need of a time evolution. Third, it is optimised for the detection of non-linear causal relations, which are common in many real-world complex systems (*Strogatz*, *2014*), but that may create problems in standard causality metrics (*Granger & Terasvirta*, *1993*).

## METHODS

Suppose two vectors of elements $\mathcal{B} = \{b_i\}$ and $\mathcal{C} = \{c_i\}$ of equal size. The two elements of each pair $(b_i, c_i)$ must be related, e.g., they may correspond to the measurement of two
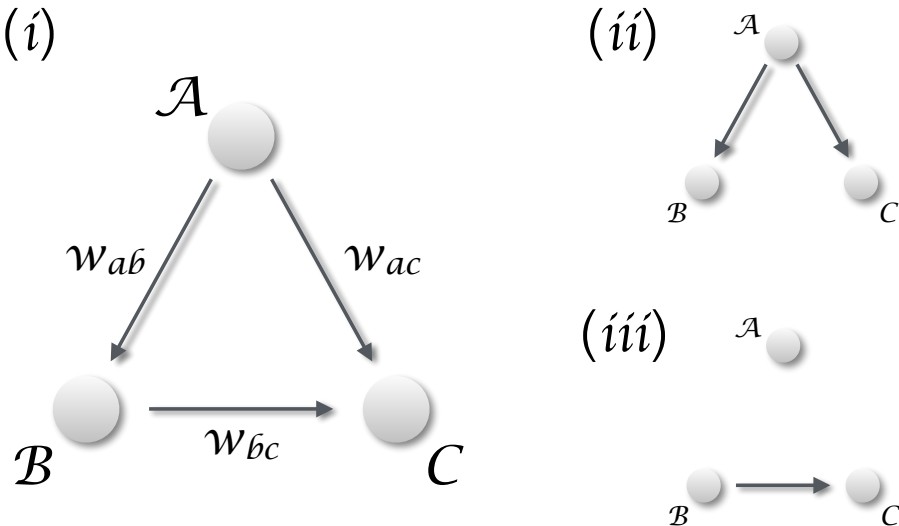

**Figure 1** **Distinguishing causality from correlation.** (i) General situation, in which three elements $\mathcal{A}$, $\mathcal{B}$ and $\mathcal{C}$ interact in a simple triangular configuration. If one is interested in the relation between $\mathcal{B}$ and $\mathcal{C}$, two different scenarios may arise. (ii) When $\mathcal{A}$ is dominating the dynamics, any common dynamics between $\mathcal{B}$ and $\mathcal{C}$ will be a correlation, generated by the external confounding factor. (iii) The situation corresponding to a real causality between $\mathcal{B}$ and $\mathcal{C}$.

biomarkers in a same subject. In the case of $\mathcal{B}$ and $\mathcal{C}$ being time series, clearly $(b_i, c_i)$ would correspond to measurements at time $i$; yet, as already introduced, such dynamical approach is not required.

Starting from these vectors, some of their elements are labelled as *extreme* when they exceed a threshold, i.e., $b_i > \tau_b$ and $c_i > \tau_c$. If a causality relation is present between them, such that $\mathcal{B} \to \mathcal{C}$, this should affect the way extreme events appear. First, under non-extreme dynamics, the two systems $\mathcal{B}$ and $\mathcal{C}$ are loosely coupled. Especially when the relation is of a non-linear nature, small values in the former system are dampened during the transmission. Second, most of the extreme values of $\mathcal{B}$ should correspond to extreme values of $\mathcal{C}$, as extreme signals will be amplified from the former to the latter by the non-linear coupling. Third, extreme values of $\mathcal{C}$ only partially correspond to extreme values of $\mathcal{B}$; due to its internal dynamics, $\mathcal{C}$ can display extreme events not triggered by the other element. An example of these three rules is depicted in Fig. 2; note how extreme events (red bars) in $\mathcal{B}$ always propagate to $\mathcal{C}$, while the second extreme event of $\mathcal{C}$ is caused by its internal dynamics and is not propagated.

Let us denote by $p_1$ the probability that an extreme event in $\mathcal{C}$ also corresponds to an extreme event in $\mathcal{B}$, i.e., $p_1 = P(b_i > \tau_b | c_i > \tau_c)$. Conversely, $p_2$ will denote the probability that an extreme event in $\mathcal{B}$ corresponds to an extreme event in $\mathcal{C}$, i.e., $p_2 = P(c_i > \tau_c | b_i > \tau_b)$. In the case of a real causality, the second condition implies that $p_1 \approx 1$, the third one that $p_2 \ll 1$. On the other hand, in the case of an external confounding effect, and if the two thresholds are chosen such that the probability of finding extreme events is the same for both elements, it is easy to see that $p_1 \approx p_2$. Notice that the same is true if $\mathcal{B}$ and $\mathcal{C}$ are bidirectionally interacting.
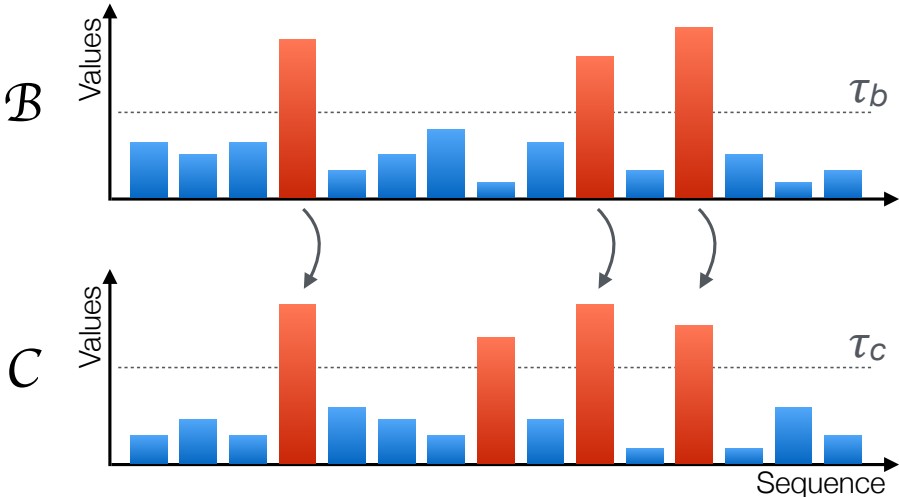

**Figure 2 Graphical representation of the proposed metric.** A system $\mathcal{B}$ is causing another system $\mathcal{C}$ when extreme values in the former, represented by red bars, propagate to the second; the opposite may nevertheless not happen, as $\mathcal{C}$ can also generate extreme values due to its own internal dynamics. The horizontal axis represents sequences of observations, but not necessarily a time evolution.

The previous analysis suggests that the necessary condition for having a $\mathcal{B} \rightarrow \mathcal{C}$ causality is $p_1 > p_2$. The statistical significance can be quantified through a binomial two-proportion $z$-test:

$$z = \frac{p_1 - p_2}{\sqrt{\hat{p}(1-\hat{p})(\frac{1}{n_1} + \frac{1}{n_2})}}, \tag{1}$$

with $n_1$ and $n_2$ the number of events associated to $p_1$ and $p_2$, and $\hat{p} = (n_1 p_1 + n_2 p_2)/(n_1 + n_2)$. The corresponding $p$-value can be obtained through a Gaussian cumulative distribution function.

Before demonstrating the effectiveness of the proposed causality metric, it is worth discussing several aspects of the same.

First of all, the attentive reader will notice the similarity of this method with some metrics for assessing synchronisation in time series. For instance, local maxima and their statistics were considered in *Quiroga, Kreuz & Grassberger* (*2002*), and event coincidences in *Donges et al.* (*2015*). In both cases, an essential ingredient is the time evolution: extreme events in one time series are identified and related to those appearing in a second time series, and the delay required for their transmission assessed through a time shift optimisation. While this yields an estimation of the direction of the information flow between two time series, it cannot be applied to systems whose time evolutions are not accessible. The metric here proposed has the advantage that can be applied to static data sets, in principle paving the way to the construction of data mining algorithms based on causality. This applicability to static data sets is also the main difference with respect to *Gómez-Herrero et al.* (*2015*), which proposes a method for the detection of relations between large ensembles of short time series. While *Gómez-Herrero et al.* (*2015*) allows analysing *fast* systems, described by time

series comprising only a handful of values, it is still not applicable to static measurements, as for instance those found in genetics.

Second, the metric definition requires setting two thresholds, i.e., $\tau_b$ and $\tau_c$. This can be done using *a priori* information, e.g., when a level is accepted as abnormal for a given biomarker; or by simply explore all the parameters space, in order to assess the values of $(\tau_b, \tau_c)$ corresponding to the lowest *p*-value. This may result especially useful in those situations for which the input elements are not well characterised: beyond the identification of causality relations, this method may also be used to define what an abnormal value is. Additionally, the form of detecting extreme events through a threshold is different from similar approached in the literature. For instance, *Quiroga, Kreuz & Grassberger* (*2002*) defines the events of interest as local maxima, independently of their amplitude; some of these events may not pass the threshold filtering here proposed, which only considers extreme (in the sense of *not normal* or *not expected*, but not necessarily of *maximal*) values.

Third, we have previously stated that the presence of a confounding effect can be correctly detected, and that in such situations the metric would not detect a statistically significant causality. According to the Common Cause Principle (*Pearl*, *2003*), two variables are unconfounded *iff* they have no common ancestor in the causal diagram; and ensuring this requires including the confounding effects in the analysis, i.e., detect if there are causalities $\mathcal{A} \to \mathcal{B}$ and $\mathcal{A} \to \mathcal{C}$ in the diagram of Fig. 1. In the context here analysed, a confounding effect would be detected as the presence of co-occurring extreme events, generated by the confounding element, in both vectors of data. This requires the confounding element to influence in the same way both analysed elements, or, in other words, to have the same coupling strength between $\mathcal{A} \to \mathcal{B}$ and $\mathcal{A} \to \mathcal{C}$. Additionally, if the causality $\mathcal{B} \to \mathcal{C}$ is mixed with an external influence, the latter cannot be detected if the strength of the former is greater—that is, a strong causality can mask a confounding effect. For all this, the proposed method does not always allow to discriminate true causalities from spurious relationships, although it provides important clues about which one of these two effects is having the strongest impact.

## RESULTS

We first test the proposed metric with synthetic data. Figure 3 presents the evolution of the *p*-value for two vectors $\mathcal{B}$ and $\mathcal{C}$, whose values are drawn from different distributions. Two situations are compared. First, a real $\mathcal{B} \to \mathcal{C}$ causality, such that $c_i = c_i + \gamma b_i^n$ (n being the order of the coupling)—solid lines in Fig. 3. Second, a confounding effect in which $b_i = b_i + \gamma a_i^n$ and $c_i = c_i + \gamma a_i^n$—dashed lines in Fig. 3. It can be appreciated that the *p*-values of real causalities drop to zero with small values of coupling constants; and that non-linear couplings perform better than linear ones. When the same analysis is performed using other causality standard metrics, such clear behaviour is not observed. Specifically, Fig. 4 presents the evolution of the *p*-value, as obtained for Gaussian distributions by the Granger Causality and the Transfer Entropy. The former metric rejects, for all coupling constants, the presence of a causality. As for the Transfer Entropy, it correctly detects the presence of a relationship, but only for very high coupling constants; additionally, it is not able to detect

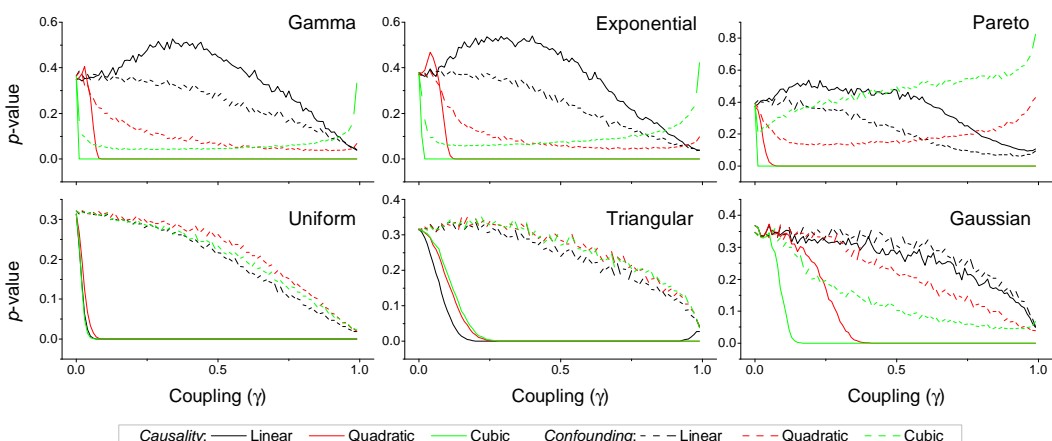

**Figure 3** *p*-value obtained by the proposed causality metric, for vectors of synthetic data drawn from six different distributions, as a function of the coupling constant *γ*—see main text for details. Black, red and green lines respectively correspond to linear, quadratic and cubic couplings; solid lines depict true causalities (as in Fig. 1(ii)), dashed lines spurious ones (Fig. 1(iii)). Each point corresponds to 10, 000 realisations.

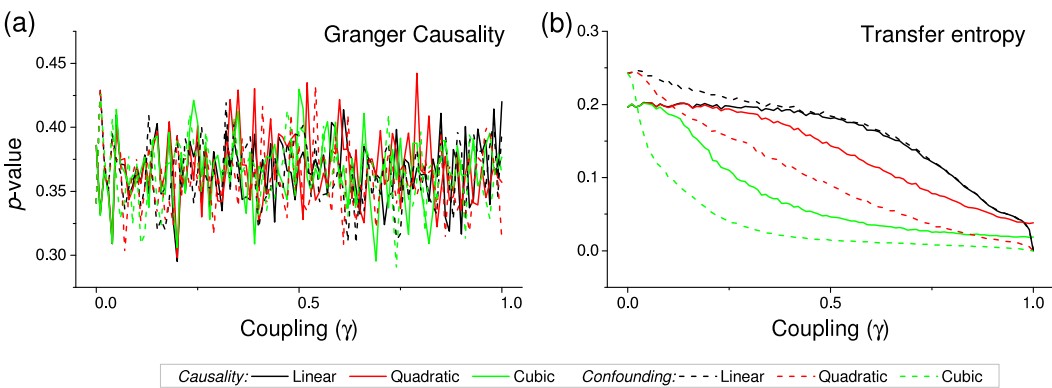

**Figure 4** *p*-value obtained by two standard causality metrics, for vectors of synthetic data drawn from Gaussian distributions, as a function of the coupling constant *γ*. (A) Corresponds to the Granger Causality, (B) to the Transfer Entropy. Black, red and green lines respectively correspond to linear, quadratic and cubic couplings; solid lines depict true causalities (as in Fig. 1(iii)), dashed lines spurious ones (Fig. 1(ii)).

the presence of confounding effects—note that the three dashed lines in Fig. 4B are almost always below the corresponding solid ones. In some cases, a confounding effect, especially when highly non-linear, can foul the proposed metric and yield a low *p*-value—see, for instance, the cubic confounding coupling for a gamma distribution in Fig. 3. Such situations can easily be identified by comparing the *p*-values for $\mathcal{B} \rightarrow \mathcal{C}$ and $\mathcal{C} \rightarrow \mathcal{B}$: in the case of a true causality, which is by definition directed, the *p*-value should be small only for one of them. An example of this is depicted in Fig. 5, which shows the evolution of the *p*-values for a confounding effect (Fig. 5A) and a causality (Fig. 5B), for vectors of Gamma distributed values. Once the limitations and requirements about confounding effects, as defined in the previous section, are taken into account, discriminating between true and spurious

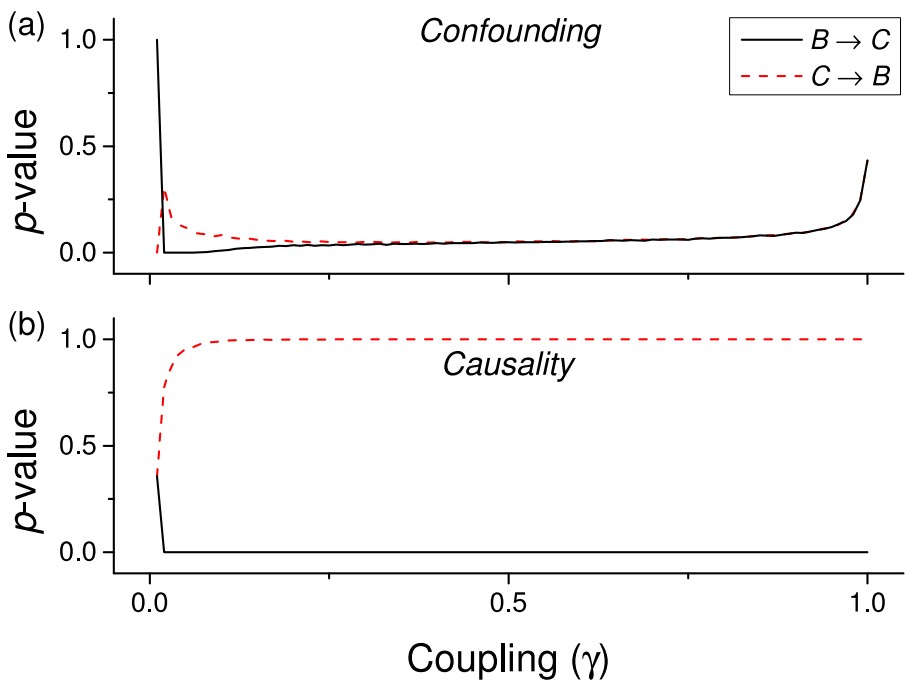

**Figure 5** Evolution of the *p*-value of the causality, when considering both $\mathcal{B} \to \mathcal{C}$ and $\mathcal{C} \to \mathcal{B}$ tests for a cubic coupling and for data drawn from a Gamma distribution (as in green lines of the first panel of Fig. 3. (A) Reports the results for a confounding effect, (B) for a true causality between $\mathcal{B}$ and $\mathcal{C}$.

causalities only requires calculating the two opposite *p*-values, and checking whether they are both small.

The necessity of detecting extreme events introduces a drawback in the method, i.e., the need of having a large set of input values to reach a stable statistics. This problem is explored in Fig. 6, which depicts the *p*-value obtained as a function of the number of input values. Depending on the kind of relation to be detected, between 2 and 4 thousand values are required.

One of the advantages of the proposed metric is that it can be applied both to cross-sectional and longitudinal data. In other words, the metric can be used to study both those systems that do not present a temporal evolution, but for which information corresponding to different instances is available; and those systems whose evolution through time can be observed. Here we show such flexibility in the detection of the causality between two noisy Kuramoto oscillators (*Kuramoto*, *2012*; *Rodrigues et al.*, *2016*). Suppose two oscillators whose phases are defined as:

$$\dot{\phi}_{\mathcal{B}} = \kappa_{\mathcal{B}} + \xi \tag{2}$$

$$\dot{\phi}_{\mathcal{C}} = \kappa_{\mathcal{C}} + \gamma \sin(\phi_{\mathcal{B}} - \phi_{\mathcal{C}}) + \xi. \tag{3}$$

$\kappa$ is the natural frequency of each oscillator ($\kappa_{\mathcal{B}} \neq \kappa_{\mathcal{C}}$), and $\xi$ an external uniform noise source. The coupling constant $\gamma$ defines the way the two oscillators interact, with independent dynamics for $\gamma \approx 0$, and a causality $\phi_{\mathcal{B}} \to \phi_{\mathcal{C}}$ for $\gamma > 0$. The longitudinal causality can be detected by considering the time series created by $\dot{\phi}_{\mathcal{B}}$ and $\dot{\phi}_{\mathcal{C}}$, thus focusing

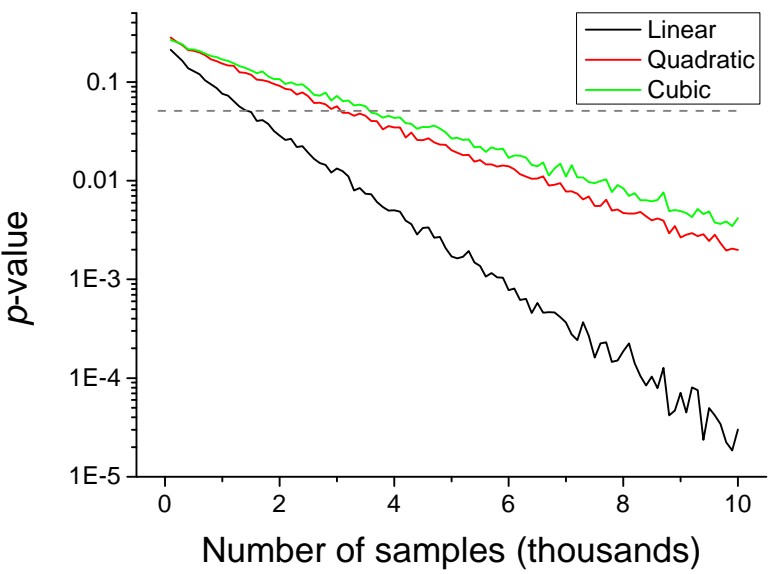

**Figure 6** **Evolution of the $p$-value of the causality, for a triangular distribution, as a function of the number of values included in the input vectors.** Black, red and green lines respectively correspond to linear, quadratic and cubic couplings.

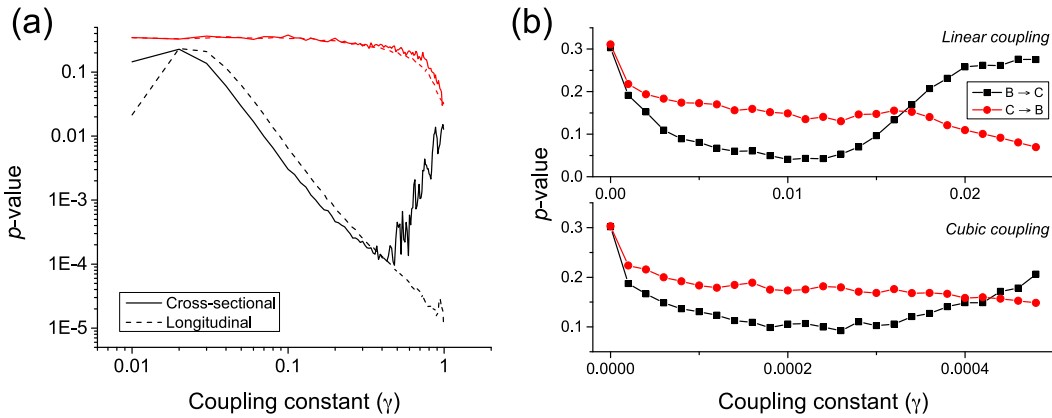

**Figure 7** **(A) Evolution of the $p$-value of the causality test between two Kuramoto oscillators, for different values of the coupling constant $\gamma$.** Solid and dashed lines respectively correspond to a cross-sectional and longitudinal study—see main text for details. Black lines correspond to the proposed metric, red ones to Granger Causality. (B) $p$-value for two coupled Rössler oscillators as a function of the coupling constant $\gamma$, for a linear and cubic coupling.

on how abnormal *jumps* in the phase of the oscillators is transmitted from the former to the latter. The $p$-value of the metric is represented in Fig. 7A by the black dashed line. The equivalent cross-sectional analysis requires multiple realisations of the previous dynamics; for each one of them, one single pair of values $(\dot{\phi}_{\mathcal{B}}, \dot{\phi}_{\mathcal{C}})$ is extracted, corresponding to the largest variation of $\phi_{\mathcal{B}}$, and thus to the most extreme jump in the phase of the first oscillator. The evolution of the corresponding $p$-value is shown in Fig. 7A by the black solid line.

Both the longitudinal and cross-sectional analyses yield similar results, suggesting that dynamical and static causalities are equivalent under the proposed metric. Only when the coupling is large, i.e., above 0.5, the longitudinal (i.e., time based) analysis yields better $p$-values than the cross-sectional one, as the latter is probably confounded by the presence of a strong correlation. Figure 7A further depicts the behaviour of the $p$-value when calculated using the Granger Causality metric; it can be appreciated that the proposed causality metric is more sensitive, especially for small coupling constants.

An important characteristic of complex systems is that their constituting elements usually have a chaotic dynamics (*Strogatz, 2014*), making more complicated the task of detecting causality between them. We here test the proposed metrics by considering two unidirectionally coupled Rössler oscillators ($\mathcal{B} \rightarrow \mathcal{C}$) in their chaotic regime—see (*Rulkov et al.* (*1995*) for details. We consider both linear and cubic couplings; following the notation in *Rulkov et al.* (*1995*), this means:

$$\dot{y}_1 = -(y_2 + y_3) - \gamma(y_1 - x_1), \quad \text{and} \tag{4}$$

$$\dot{y}_1 = -(y_2 + y_3) - \gamma(y_1 - x_1)^3. \tag{5}$$

Time series are created by sampling the second dimension of each oscillator (i.e., $x_2$ and $y_2$) with a resolution lower than the intrinsic frequency. Figure 7B depicts the evolution of the $p$-value for low coupling strengths $\gamma$, thus ensuring that the system is *generalised synchronised*. For $\gamma \approx 0.01$ ($\gamma \approx 2 \cdot 10^{-4}$ for cubic coupling), a true causality is detected, while for $\gamma > 0.015$ ($\gamma > 4 \cdot 10^{-4}$) the two oscillators start to synchronise.

The possibility of combining a cross-sectional analysis of extreme values with a longitudinal analysis opens new doors towards the understanding of systems for which both aspects can be studied at the same time. Here we show how this can be achieved in the analysis of functional networks representing the structure of brain activity in healthy subjects (*Bullmore & Sporns, 2009*; *Rubinov & Sporns, 2010*). The data set corresponds to electroencephalographic (EEG) recordings of 40 subjects during 50 trials of an object recognition task (details can be found in *Zhang et al.* (*1995*) and references within), obtained through the UCI KDD archive (*Bay et al., 2000*). For each trial and subject, 19 time series of 256 samples were available, corresponding to one second of recording of 19 EEG channels in the 10–20 configuration. The longitudinal analysis was performed by calculating the causality using the raw time series. On the other hand, the cross-sectional analysis relies on identifying the propagation of extreme events, as in the case of the Kuramoto oscillators. Extreme events are defined as those for which the energy of the signal is maximum in a given time series; the energy is defined, at each time point, as the deviation with respect to the mean, normalised by the standard deviation of the signal—i.e., as the absolute value of the Z-Score.

Figure 8A depicts a box plot of the proportion of significant pairs of channels (i.e., pairs of channels for which a causality was detected), in both the cross-sectional (blue) and longitudinal (red) analyses, for different significance levels $\alpha$. In the case of the cross-sectional analysis, each value corresponds to the results for a single subject. Results are qualitatively equivalent, with the longitudinal analysis detecting slightly less links than the cross-sectional one for small values of $\alpha$. Figures 8B and 8C depict the 10 most

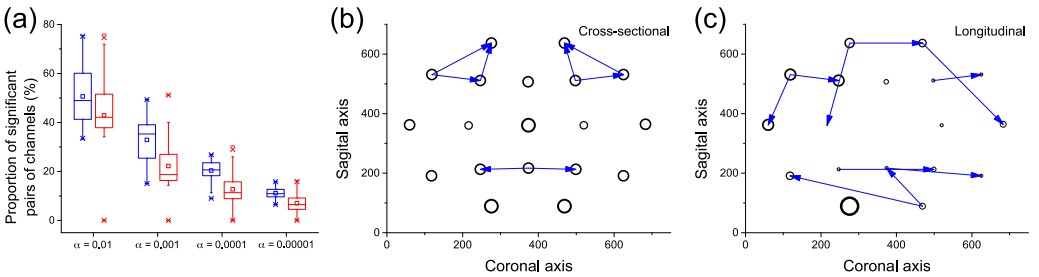

**Figure 8** **Analysis of causality in EEG data.** (A) Proportion of pairs of channels in which causality has been detected, for cross-sectional (blue) and longitudinal (red) analyses, as a function of the significance level $\alpha$. (B) Top-10 causality links in the cross-sectional analysis. (C) Top-10 causality links in the longitudinal analysis. In (B) and (C), the size of each node is proportional to its number of connections (i.e., its degree of participation in the cognitive task).

significant links, as detected by both analyses. While not completely equivalent, both graphs suggest that some areas are identified as active by both methods, e.g., the frontal lobe on the top and the visual and somatosensory integration area in the bottom. Remarkably, these two regions are expected to be relevant for the task studied, i.e., object identification: the former for higher function planning, i.e., react to the image shown, the latter in the processing of visual inputs.

## CONCLUSIONS AND DISCUSSION

In conclusion, we presented a novel metric able to detect causality relationships both in static and time-evolving data sets, thus overcoming the limitation of existing metrics that rely on time series analysis. The proposed metric is designed to detect the propagation of extreme events, or shocks, and as such is more efficient when non-linear relations are present; it is further able to discriminate real from spurious causalities, thus enabling the detection of confounding effects. The effectiveness of the metric has been tested through synthetic data; data obtained from simple and chaotic dynamical systems, i.e., Kuramoto and Rössler oscillators; and through EEG data representing the activity of the human brain during an object recognition task.

In spite of the advantages that the proposed metric presents, and that have been described throughout the text, two limitations have to be highlighted. First, the reduced sensitivity of the metric to linear causality relationships, and in the analysis of data without long tail distributions, i.e., without clear extreme events—see Fig. 3 for further details. Second, the need of large quantities of data, in the order of several thousands of observations, to reach statistically significant results (Fig. 6).

The possibility of detecting causality in static data sets is expected to be of increasing importance in those research fields in which time dynamics are not available, and that require ensuring that a causality is not just the result of the presence of a confounding factor. For instance, one may considering the raising field of biomedical data analysis (*Prather et al.*, *1997*; *Cios & Moore*, *2002*; *Han*, *2002*). The custom solution is to resort to data mining algorithms, which allow to detect and make explicit patterns in the input data, with the final objective of using such patterns in diagnostic and prognostic models (*Vapnik*, *2013*).

Nevertheless, data mining (and machine learning in general) is based on the Bayes theorem, a form of statistics of co-occurrences, and thus on a generalised concept of correlation. These methods are thus sensitive to the confounding effects that are frequently in place, as genes and metabolites create an intricate network of interactions. Resorting to classical causality metrics, like Granger's one, is not possible, as time series are seldom available—measuring gene expression or metabolite levels is an expensive and slow process. In spite of this, causality is an essential element to be detected: if one only focuses on correlations, there is a risk of detecting elements whose manipulation does not guarantee the expected results on the system (*Salmon et al.*, *2000*; *Cardon & Palmer*, *2003*; *Vakorin, Krakovska & McIntosh*, *2009*). We foresee that the proposed causality metric can be an initial solution to this problem, by providing a causality test that can be applied to static data, and that could be used as the foundation of a new class of data mining algorithms.

A Python implementation of the proposed causality metric is freely available at www.mzanin.com/Causality.

### Funding
The author received no funding for this work.

### Competing Interests
Massimiliano Zanin is an Academic Editor for PeerJ and PeerJ Computer Science.

### Author Contributions
- Massimiliano Zanin conceived and designed the experiments, performed the experiments, analyzed the data, wrote the paper, prepared figures and/or tables, reviewed drafts of the paper.

### Data Availability
The data set used was downloaded from the UCI KDD archive at:
https://archive.ics.uci.edu/ml/datasets/EEG+Database.

Additionally, the source code for calculating the metrics can be downloaded from:
http://www.mzanin.com/Causality/.

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
