# Peer review of "On causality of extreme events"

_PeerJ, doi:10.7717/peerj.2111_

## Round 0.1 · original submission · Major Revisions

All three referees did a fantastic job. I hope you find the comments and suggestions useful and elect to resubmit. I agree with suggestions such as these "in particular the authors must discuss the sensitivity of their method to the computation of the probabilities. For example how the choice of the binning size or other approach to compute the probabilities affect the results." and also a general expansion of methodology.

Reviewer 1 ·

Basic reporting

The introduction does a good job of summarizing the discussion on causality, but overall I found it very disorganized. It introduces many different concepts and ideas very quickly and seems to be a bit unstructured. I think this can be resolved by generating more structured paragraphs. Have a clear topic sentence that defines a detailed paragraph. Having 2 or 3 sentences in a paragraph potentially leaves off key information for readers that are not as well versed in this literature. For instance, when bringing up the genetic examples, it would be nice to re-link how that example fits in the content of the paragraph.
In general, I think your conclusion is nicely structured and should represent a format more similar to that.
Avoid putting text in brackets when it often can be left in the sentence and separated by commas. For example: Line 30. “but not necessarily” does not need to be in brackets
Figure 1 should also have clear indications to separate each proportion of it. Perhaps, label figures with Roman numeral to make it easier to distinguish than “left bottom, right bottom, etc”.

Experimental design

No comments

Validity of the findings

No comments

Additional comments

The article introduces a novel metric to detect causality when time series data is not available. I find this interested and a useful contribution to science. It would be interesting to have seen the data tested on a real data set rather than synthetic, although I understand the limitations behind it. I look forward to seeing it used in additional contexts.

Reviewer 2 ·

Basic reporting

In their works, the authors propose a methodology to detect both linear and non-linear causalities across both cross-sectional and longitudinal data sets and to differentiate between true causalities from false correlations. They validate their method using simulated data and experimental neurological data.
The merit of the paper is to address the interactions between two systems through a simple and original approach. This is relevant to study the interactions between static and non-linear systems whereas traditional methods such as Granger’s causality cannot be applied.
The authors do not compare their method to other existing methods applied to the same problem.
Overall, we found the paper well written with a good discussion of the methodology used. We believe that upon addressing our main concerns, the work proposed by the authors is suitable for publication.

Experimental design

While we find the methodology presented in the paper to be well-suited for static data, the authors do not refer enough to the other methods such as data mining and machine learning, Granger causality, cointegration, or transfer entropy which are simply quickly mentioned in the introduction. I believe that the methodology proposed might benefit from the vast literature in detecting coupling in time series using these techniques. This work need to be enriched with salient literatures providing more sound evidence to the authors’ comments. In particular, each approach must be detailed with appropriate citations and the authors must provides the advantages and limitations of each of them.

Validity of the findings

In their work, the authors provide tangible pros of their method but failed to provide the cons of their approach in a clear and comprehensible language either in the method or in the discussion section. The novel approach should be tested against other methods in solving the same problem. From the comparison of the performance of other methods compare to their, the authors can derived constructive and tangible conclusions. In particular the authors must discuss the sensitivity of their method to the computation of the probabilities. For example how the choice of the binning size or other approach to compute the probabilities affect the results.

Additional comments

In the figures, the numbers are formatted according to the European style which might confusing with regard to the American style of decimal notations. This issue should be fixed with respect to the broad audience of the journal. Also, we found a typo in line 55 where p1 should read p2 instead.

Reviewer 3 ·

Basic reporting

No Comments

Experimental design

No Comments

Validity of the findings

No Comments

Additional comments

This paper describes a new metric that can be used to infer causality between two datasets. The metric can be used on datasets that do not have temporal information attached and can therefore be used in a wider variety of setups. The metric is defined, analyzed, and used on synthetic as well as real datasets.

The paper is clean, clear, and well-written. The author has presented the metric and analyzed its properties well. However, as a general comment, given the results that the metric underperforms on linear datasets (fig. 2), and requires considerable samples (fig. 3), the abstract can be revised to include such caveats. Also, the meaning of cross-section and longitudinal may not be standard, so I suggest using different terminology or phrase.

More specifically:

- It looks like the cross-sectional observations are similar to an ensemble of multiple short processes (not necessarily arranged in time). Could the author compare his work with [a] which, though it uses transfer entropy, is able to resolve information transfer through an ensemble of independent reputations.
- Temporal information if available should only aid in inferring causality. From reading the paper and going through the examples (fig. 6), it feels as if the paper claims that even if temporal information is available, computing it using the new metric will give better results. why?
- This also relates to the thresholds for selecting extreme events. In the longitudinal case, it appears that the approach is no different from reference [15]. It would be nice to see a comparison, in words, or through examples (I believe the authors for that paper have also provided a MATLAB code for executing the same)
- The problem of data size is critical here, especially for thresholds that may be high, or there is too much noise in the data. Kindly discuss that aspect in your conclusion
- Minor fixes: p1 in line 55 should be p2; Fig. 2 caption should say 10,000 realizations; arrows not showing in Fig. 6 for the directed links; kindly define ‘weight’ of a node



[a] G. Go ́mez-Herrero, W. Wu, K. Rutanen, M. C. Soriano, G. Pipa, and R. Vicente, Assessing coupling dynamics from an ensemble of time series, Entropy 17, 1958 (2015)

---

## Round 0.2 · accepted · Accept

Thank you for taking the time to thoroughly revise. I love it.